# On Hardware-efficient Inference in Probabilistic Circuits

**Lingyun Yao**[1]    **Martin Trapp**[2]    **Jelin Leslin**[1]    **Gaurav Singh**[1]    **Peng Zhang**[1]    **Karthekeyan Periasamy**[1]

**Martin Andraud**[1,3]

[1]Department of Electronics and Nanoengineering, Aalto University, Espoo, Finland
[2]Department of Computer Science, Aalto University, Espoo, Finland
[3]ICTEAM, UCLouvain, Louvain-La-Neuve, Belgium

## Abstract

Probabilistic circuits (PCs) offer a promising avenue to perform embedded reasoning under uncertainty. They support efficient and exact computation of various probabilistic inference tasks by design. Hence, hardware-efficient computation of PCs is highly interesting for edge computing applications. As computations in PCs are based on arithmetic with probability values, they are typically performed in the log domain to avoid underflow. Unfortunately, performing the log operation on hardware is costly. Hence, prior work has focused on computations in the linear domain, resulting in high resolution and energy requirements. This work proposes the first dedicated approximate computing framework for PCs that allows for low-resolution logarithm computations. We leverage Addition As Int, resulting in linear PC computation with simple hardware elements. Further, we provide a theoretical approximation error analysis and present an error compensation mechanism. Empirically, our method obtains up to $357\times$ and $649\times$ energy reduction on custom hardware for evidence and MAP queries respectively with little or no computational error.

## 1   INTRODUCTION

The development of smart sensing and Internet-of-things applications based on embedded artificial intelligence (AI) pushes the computation of machine learning (ML) methods directly onto edge devices. On one hand, dedicated ML processors have increased the energy efficiency of deep feed-forward neural networks (NNs) by $10\times - 100\times$ compared to Graphical Processing Units [Seo et al., 2022]. On the other, NNs that have been adopted into real-world use often raise concerns related to their reliability, fairness, and inter-

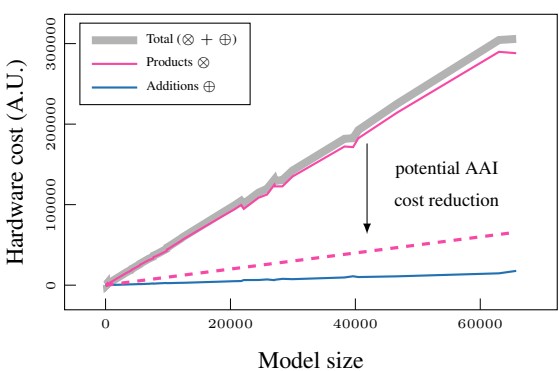

Figure 1: Potential hardware cost savings ( - - - ) for inference in probabilistic circuits through approximate computing with addition-as-int (AAI).

pretability [Marcus, 2020, Heljakka et al., 2023] alongside high inference costs [Xu et al., 2018, Strubell et al., 2019]. Moreover, NNs are generally trained on only a pre-specified task. Thus, real-world edge AI applications require an *effective* hardware acceleration of ML models that are *probabilistic*, *i.e.*, they enable reasoning in an uncertain world [Ghahramani, 2015], and *tractable*, *i.e.*, they can reliably answer many probabilistic queries without re-deployment.

Recent work on tractable probabilistic models, specifically on probabilistic circuits (PCs) [Choi et al., 2020], poses a promising avenue as these models (i) exhibit high expressive efficiency (representational power), (ii) enable reliable [Ventola et al., 2023, Peharz et al., 2019] and fair [Choi, 2022] reasoning, and (iii) allow many probabilistic queries to be computed tractably by design. Moreover, PCs can be understood as computational graphs composed of simple arithmetic operations, easily translated into hardware computations. Yet, while pioneering works have explored acceleration of PCs on hardware [Choi et al., 2022, Sommer et al., 2018, Sommer et al., 2020, Shah et al., 2021, Shah et al., 2022], the hardware acceleration of PCs still poses open challenges. In particular, their *irregularity* (*i.e.*, PCs are sparse, limiting computation parallelism [Shah et al., 2019]) and *computation resolution* (*i.e.*, arithmetic opera-

*Accepted for the 40th Conference on Uncertainty in Artificial Intelligence* (UAI 2024).

tions are performed on probabilities that can get as low as $10^{-88}$ [Sommer et al., 2020]) hinders their deployment on edge devices, where efficiency and low resolution are key.

Tackling the computation resolution challenge, we start with the following observation: in software, PC inference methods typically use a logarithm representation for the computation, alternating between logarithm multiplication and linear additions ( using the "log-sum-exp" trick to transfer the operand back to the linear domain, add them, and convert them back to logarithm), while all hardware PC accelerators prefer using a fully linear computation with higher resolution [Sommer et al., 2020, Shah et al., 2022]. Hardware limitations explain this difference. First, alternating between logarithm and linear domains would require specific hardware blocks for encoding/decoding, which would induce a higher cost and limit speed. Second, computing a PC fully in the logarithm domain is inefficient due to the prohibitive cost of logarithmic adders [Sommer et al., 2020]. Instead, a full linear computation, using floating-point or Posit formats, is preferred [Sommer et al., 2020, Shah et al., 2022]. In this case, as illustrated in Fig. 1, the hardware cost steadily increases with the model complexity to handle the increasing dynamic range of the PC. Typical PC benchmarks require an effective range of 30-40 floating-point bits [Sommer et al., 2020]. As a comparison, deep NNs typically require 5-8 integer bits for inference. This cost is heavily dominated by multiplications, *i.e.*, a floating-point multiplier consumes $6\times$ more energy than an adder in a 45 nm process technology [Olascoaga et al., 2019].

We propose a dedicated approximate computing framework to efficiently compute PCs on hardware. It is based on a similar alternation between log and linear computations, which we refer as the "exp-sum-log" trick. This computation leverages Addition As Int (AAI) [Mogami, 2020] (will be explained later in Section 3.2) to approximate the product of two variables as a log addition, relying on Mitchell's approximation [Mitchell, 1962] (will be explained later in Section 3.2) .

Our contributions can be summarized as follows:

- From theoretical foundations, we devise a dedicated strategy for the safe multiplier replacement with AAI, minimizing the accuracy loss (Section 4.3).
- We compensate AAI induced loss in accuracy through a specific error correction (Section 4.4).
- Lastly, we derive a hardware-efficient architecture for marginal and MAP inference (Section 4.5), and show through hardware simulations that the proposed approach substantially reduces computational costs. In particular, we observe savings of up to $357\times$ for marginal queries and up to $649\times$ for MAP queries, in both cases with low approximation error (Section 5).

GitHub repository avaliable: GitHub Repository.

## 2 BACKGROUND

**Notation:** We use upper case letters to denote random variables (RVs) (*e.g.*, $X$) and lower case letters for realizations of RVs (*e.g.*, $x$). Further, we use **bold** font for vectors (*e.g.*, $\boldsymbol{X}, \boldsymbol{x}$) and matrices (*e.g.*, $\boldsymbol{M}$).

**Probabilistic circuits (PCs)** [Choi, 2022] are a unifying framework of existing tractable models (*e.g.*, [Darwiche, 2003, Poon and Domingos, 2011, Rahman et al., 2014, Kisa et al., 2014]). They provide a concise language to represent and reason about tractable (exact and efficient) probabilistic inference.

Given a set of $d$ RVs $\boldsymbol{X}$, a probabilistic circuit is a function $c\colon \mathbb{R}^d \to \mathbb{R}_+$ (typically a density or mass function) represented by a parameterized computational graph $\mathcal{G}$ consisting of sum $\oplus$, product $\otimes$, and leaf units. Each computational unit is defined over a set of variables, called its *scope* [Trapp et al., 2019], and every non-leaf unit computes an algebraic operation over sub-circuits. The scope of each non-leaf unit is given by the union of the scopes of its sub-circuits (inputs). Sum units compute a weighted sum of sub-circuits, *i.e.*, $\sum_{s_i=1}^{k} w_{s_i} c_{s_i}(\boldsymbol{x})$ with $w_{s_i} \geq 0$, product units multiply sub-circuits, *i.e.*, $c_i(\boldsymbol{x}) \cdot c_j(\boldsymbol{x})$, and leaf units evaluate a tractably integrable function, *e.g.*, the indicator function $\mathbb{1}[X_u = 1]$, on their inputs. W.l.o.g. we assume that product units compute binary products and sum nodes are normalized, *i.e.*, $\sum_{s_i=1}^{k} w_{s_i} = 1$. Fig. 2(a) illustrates a PC over discrete RVs using indicators.

A particularly relevant class of PCs are *smooth* and *decomposable* circuits, as both properties are requirements for many probabilistic queries to be tractable, *i.e.*, time complexity is linear in the model size. We will briefly review the relevant properties.

**Definition 2.1** (Smooth & Decomposability). *A sum unit is **smooth** if all of its sub-circuits have the same scope, i.e.,* $\sum_{s_i=1}^{k} w_{s_i} c_{s_i}(\boldsymbol{x})$*. Further, a product unit is **decomposable** if its sub-circuits have pairwise disjoint scopes, i.e.,* $c_i(\boldsymbol{y}) \cdot c_j(\boldsymbol{z})$ *with* $\boldsymbol{Y} \cap \boldsymbol{Z} = \emptyset$*. A PC is smooth if all sum units are smooth and decomposable if all product units are decomposable.*

Another important sub-class of PCs are *deterministic* PCs, as determinism is a sufficient condition for tractable maximum-a-posteriori (MAP) inference and many relevant quantities, *e.g.*, KL-divergence, can be computed analytically.

**Definition 2.2** (Determinism). *A sum unit is **deterministic** if for every complete evidence $\boldsymbol{x}$ at most one of its inputs (sub-circuits) has a positive value. Consequently, a PC is deterministic if all sum units are deterministic.*

We refer the reader to [Vergari et al., 2021] for a detailed exposé on the structural properties of PCs and their interplay with the tractability of computations.

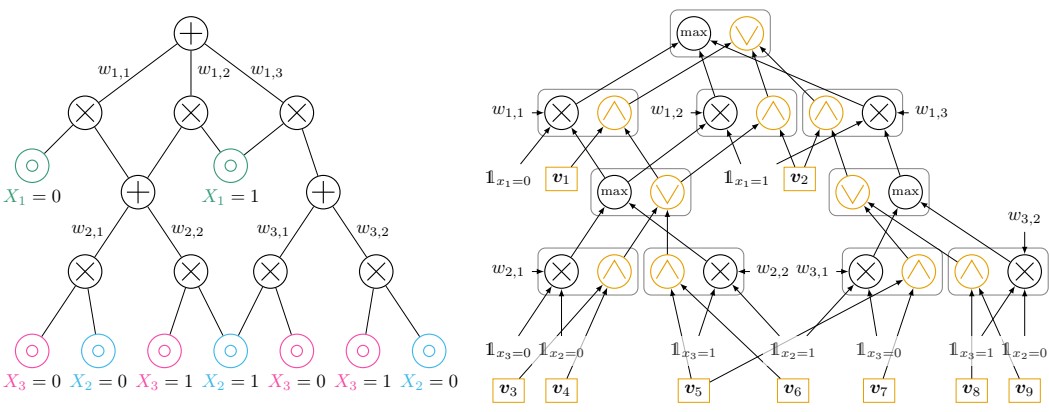

(a) Probabilistic Circuit

(b) Corresponding hardware representation for MAP Inference

Figure 2: Illustration of a PC (a) over discrete RVs ($X_1, X_2, X_3$) and the corresponding hardware realization of MAP inference (b). For this, sum nodes are replaced by max operators, and an additional propagation path for information bits ($v_i$) is added to back-track the most probable path. Arrows $\rightarrow$ indicate propagation direction.

In this work, all PCs are considered smooth and decomposable, and we present additional theoretical results for deterministic PCs. Our proposal for hardware acceleration utilizes those structural properties to perform error correction, as highlighted in Section 4.

## 2.1 PROBABILISTIC CIRCUITS ON HARDWARE

The deployment of PCs on hardware requires additional care compared to software inference: ① defining the **format and computational resolution**, ② specifying the **hardware generation process**, and ③ implementing **support for probabilistic queries**. We will briefly review common approaches for ① and ②. A dedicated PC hardware for ③ is presented in Section 4.5.

① Inferences on hardware are typically computed in the linear domain, with formats allowing for a large dynamic range, such as floating-point or posit [Sommer et al., 2020, Shah et al., 2021]. The resolution (*i.e.*, the number of exponent and mantissa bits) and energy can then be optimized depending on the PC structure [Shah et al., 2019, Sommer et al., 2020], which requires customized arithmetic blocks.

② To generate the hardware, a classical approach translates each computational unit (sum, product) into a separate hardware entity and connects multiple entities accordingly [Sommer et al., 2020, Sommer et al., 2018, Shah et al., 2019]. This approach is followed in our work, with hardware described in the hardware language Chisel. A more advanced approach maps any PC to a generic processor [Shah et al., 2022, Choi et al., 2022], containing several parallel paths (or processing elements), each computing part of the PC graph. This requires a dedicated graph compiler [Shah et al., 2021].

# 3 APPROXIMATE COMPUTING FOR PCS

Computations in PCs involve alternating between additions and multiplications. As repetitive multiplications of probability values can lead to underflow, in software, computations are typically performed in the log domain (*i.e.* model variables are represented as their logarithm). As log sums require significant computing resources, to evaluate sum units, one needs to transfer between the log to the linear domain and back. This is done using the '*log-sum-exp*' trick, to ensure numerical stability. Such trick is costly on hardware. It requires either a look-up table (LUT) [Xue et al., 2020] for the linear-log conversion, resulting in high resource consumption and slower computation, or a full computation in either domain. The log domain computation being limited by the prohibitive hardware cost of log adders [Sommer et al., 2020], PC hardware accelerators prefer the linear domain using high resolution (thus hardware cost) to avoid underflow. Instead, our proposed solution implements a similar 'trick' that we could explain as a *exp-sum-log*, where we perform the costly linear multiplication in an *approximated logarithmic form* with simple hardware, without requiring an explicit transfer into log domain. After preliminary considerations, this section details how the proposed methodology approximates a floating point multiplication, and why this is useful for PC inference.

## 3.1 PRELIMINARY CONSIDERATIONS

**P1: Floating point computation** The IEEE 754 standard floating-point format (float) uses one sign bit $S$, and sets of exponent $E$ and mantissa $M$ bits. A number $x$ is represented as $x = (-1)^{S_x} 2^{E_x - b}(1 + M_x)$, where $b$ is the exponent bias. For example, the number $x = 1/3$ represented with 16

bits (half-precision) with $b = 15$, results in:

$$\tfrac{1}{3} = (-1)^0 2^{13-b}(1 + \tfrac{341}{1024}) \tag{1}$$

$$\tfrac{1}{3} = \begin{array}{|c|c|c|c|c|c|c|c|c|c|c|c|c|c|c|c|}\hline 0 & 0 & 1 & 1 & 0 & 1 & 0 & 1 & 0 & 1 & 0 & 1 & 0 & 1 & 0 & 1 \\ \hline \end{array}$$

$$\underbrace{2^3 + 2^2 + 1}_{\text{Exponent}} \quad \underbrace{\tfrac{1}{2^2} + \cdots + \tfrac{1}{2^{10}} = \tfrac{341}{1024}}_{\text{Mantissa}}$$

Given two float numbers representing probabilities $x = 2^{E_x}(1 + M_x) \geq 0$ and $y = 2^{E_y}(1 + M_y) \geq 0$, their exact product $x \cdot y$ is given as:

$$x \cdot y = 2^{E_x + E_y}(1 + M_x)(1 + M_y) \tag{2}$$

If $M_x + M_y + M_x M_y < 1$:

$$= 2^{E_x + E_y}(1 + M_x + M_y + M_x M_y)$$

Otherwise:

$$= 2^{E_x + E_y + 1}(1 + \tfrac{1}{2}(M_x + M_y + M_x M_y - 1)).$$

This operation comprises four steps: (1) add the exponents, (2) multiply the mantissa, (3) normalize the mantissa to ensure it is smaller than one, and (4) round the final result to ensure it is encoded in the same number of bits. In particular, the term $M_x M_y$ requires a higher resolution than both original mantissa values, especially when representing small values as in PCs. See Appendix A for more detail.

**P2: Mitchell's approximation** Mitchell's approximation [Mitchell, 1962], initially intended to provide a log approximation of integer numbers encoded in binary. Consider a *N-bit* integer binary string **z**, which can be written as $\sum_{i=0}^{N-1} 2^i z_i$, where $z_i$ denotes the i$^{\text{th}}$ bit in the string. Assuming the leading one bit is at position $k$, this string can be represented as:

$$\boldsymbol{z} = \sum_{i=0}^{N-1} 2^i z_i = \underbrace{2^k(1 + \sum_{i=0}^{k-1} 2^{i-k} z_i)}_{=2^k(1+F)} \tag{3}$$

In this case, every bit at positions $(0, \ldots, k-1)$ has a negative power of two and, hence, represents a fractional part $F$ by definition. From the representation created by Eq. (3), *i.e.*, $\log_2(z) = k + \log_2(1 + F)$, Mitchell's method uses an approximation given by $\log_2(1 + F) \approx F$.

We can see the error is zero when the fractional part is zero or one and Mitchell's approximation underestimates the exact value due to the difference of $\log_2(1 + F)$ and $F$, *c.f.* Fig. 3.

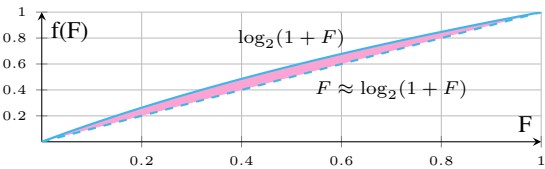

Figure 3: Input dependent approximation error ▬ introduced through Mitchell's method.

### 3.2 FROM 'LOG-SUM-EXP' TO 'EXP-SUM-LOG'

Based on the preliminary considerations, Addition as Int (AAI) [Mogami, 2020] extends Mitchell's method to floating-point multiplication. Using Eq. (2) from Section 3.1, the log computation of the product in float is given as: $\log_2(x \cdot y) =$

$$E_x + E_y + \log_2(1 + M_x) + \log_2(1 + M_y). \tag{4}$$

With no approximation, computing this would require the transfer of $(1+M_x)$ and $(1+M_y)$ from linear to log domain, introducing a high hardware cost if done exactly. Instead, AAI applies Mitchell's approximation, *i.e.* $\log_2(1 + M)$ by $\log_2(1 + M) \approx M$ (see Eq. (3) in P1). Eq. (4) becomes:

$$\log_2(x \cdot y) \approx E_x + E_y + M_x + M_y \tag{5}$$

Thus, AAI implicitly interprets the multiplication as a logarithm addition ('exp-sum-log') in a hardware-efficient way, requiring only one integer addition. This is particularly interesting for PCs where multiplication clearly dominate the hardware cost (c.f. Fig. 1).

## 4 HARDWARE-EFFICIENT INFERENCE

In principle, Section 3 showed that AAI can readily replace float multipliers used in existing PC accelerators such as [Sommer et al., 2020, Shah et al., 2019]. Yet, the AAI approximation can quickly impact the model's accuracy, as one AAI block can have up to 25% approximation error. It is shown in Fig. 6 for evidence queries on various benchmarks (see Section 5) that randomly replacing even a small percentage of multipliers can significantly increase the error, in turn limiting energy savings.

This section starts with the theoretical analysis of evidence queries (Section 4.1) and MAP queries (Section 4.2). Later, our framework mitigates the introduced error in two dedicated ways: safe multiplier replacement (Section 4.3) and error correction (Section 4.4), detailed in this section.

### 4.1 THEORETICAL ANALYSIS OF EVIDENCE QUERIES

When leveraging approximate computations in probabilistic reasoning models, it is important to understand the overall

impact of the approximation and characterize the introduced error. Therefore, we will assess the approximation error for evidence (or marginal under complete evidence, MAR) and maximum-a-posteriori (MAP) queries. We are interested in computing the expected *loss of information* when using AAI to approximate the exact multiplication. For this, we leverage the KL-divergence between the exact PC ($c$) and the circuit approximated through AAI ($\widetilde{c}$), *i.e.*,

$$\mathrm{D}_{\mathrm{KL}}\left(c||\widetilde{c}\right) = \int_{\boldsymbol{x}\in\mathcal{X}} c(\boldsymbol{x})\big(\log c(\boldsymbol{x}) - \log \widetilde{c}(\boldsymbol{x})\big)\mathrm{d}\boldsymbol{x}. \quad (6)$$

We note that the AAI approximated circuit does not have to integrate to one as AAI underestimates the exact multiplication result and our error correction may be understood as an implicit renormalization.

**Deterministic Circuits** If the circuit to approximate is deterministic (*c.f.*, Definition 2.2), the KL divergence from $\widetilde{c}$ to $c$ is available in closed-form [Vergari et al., 2021]. Leveraging this, we obtain a closed-form difference to AAI:

$$\mathrm{D}_{\mathrm{KL}}\left(c||\widetilde{c}\right) = \int c(\boldsymbol{x}) \sum_{w_i\in\mathcal{T}(\boldsymbol{x})} \Delta_{w_i}\mathrm{d}\boldsymbol{x}, \quad (7)$$

where $\Delta_{w_i}$ is the log-space approximation error of AAI for the $i^{\mathrm{th}}$ weight of the sub-tree $\mathcal{T}(\boldsymbol{x})$, *i.e.*,

$$\log p(\mathcal{T}(\boldsymbol{x})) - \log \widetilde{p}(\mathcal{T}(\boldsymbol{x}))$$
$$\propto \sum_{w_i\in\mathcal{T}(\boldsymbol{x})} \underbrace{\log_2(1 + M_{w_i}) - M_{w_i}}_{=\Delta_{w_i}}, \quad (8)$$

where we assume log-base 2 in the last step. Consequently, we obtain the approximation error of:

$$\mathrm{D}_{\mathrm{KL}}\left(c||\widetilde{c}\right) = \Delta_{\mathrm{det}} = \sum_{w_i\in c} \Delta_{w_i} \sum_{\substack{\mathcal{T}_t\in c \\ w_i\in\mathcal{T}_t}} p(\mathcal{T}_t), \quad (9)$$

as the integral in Eq. (8) simplifies to a finite sum. Eq. (9) can be computed efficiently through a bottom-up path. The full derivation is given in Appendix B.

**Non-Deterministic Circuits** In many application scenarios, the interest is in non-deterministic circuits (*e.g.*, [Poon and Domingos, 2011, Gens and Pedro, 2013, Trapp et al., 2019]). However, computing the KL divergence in this case is not possible analytically. Therefore, we present an approximation on the KL divergence that allows the estimation of the error induced by AAI, denoted as $\Delta_{\mathrm{dc}}$. For this, we use the fact that $\log(x + y) \approx \log(x) + y/x$ and after some algebraic manipulations, we obtain an approximation on the KL given by: $\Delta_{\mathrm{dc}} \approx$

$$\int c(\boldsymbol{x}) \left( \sum_{w_i\in\mathcal{T}_{\sigma_{\boldsymbol{x}}(1)}} \Delta_{w_i} - \sum_{j=2}^{k} \widetilde{p}(\mathcal{T}_{\sigma_{\boldsymbol{x}}(j)}) \right) \mathrm{d}\boldsymbol{x} + C, \quad (10)$$

where $\Delta_{w_i}$ again denotes the $i^{\mathrm{th}}$ log-space approximation error of AAI for the respective sub-tree. Computing Eq. (10) is still intractable and requires either Monte-Carlo integration or the use of Eq. (9), which bounds Eq. (10). A detailed derivation is given in Appendix B.

## 4.2 MAP QUERIES

When performing MAP queries, we are interested in estimating the sensitivity of MAP inference to approximate computing. For this, we examine whether the most probable path in the PC stays the same when AAI replaces exact multipliers.

First, recall that in the case of MAP inference, sum units are replaced by max operations [Choi, 2022]. Our analysis is based on a max operation over two sub-circuits, denoted as left ($l$) and right ($r$) branches respectively. Each branch/sub-circuit is a binary product over inputs denoted as $x_l, x_r$ and $y_l, y_r$ respectively. Let $\Delta_E$ denote the difference in exponent values for both paths, *i.e.*, $\Delta_E = E_{x_l} + E_{y_l} - (E_{x_r} + E_{y_r})$, where $E$ denotes the exponent. Further, $\Delta_M$ and $\Delta_{M'}$ are the differences in mantissa values for both paths for exact ($\Delta_M$) and AAI ($\Delta_{M'}$) computations, respectively. Note that only the mantissa calculation is affected by AAI (as shown by Eq. (4) and Eq. (5)).

We identify the following two failure cases: (i) the case that $\Delta_E = 0$ and $\Delta_M\Delta_{M'} \leq 0$, (ii) the case that $\Delta_E \geq 0$ and $(1 + \Delta_M/\Delta_E)(1 + \Delta_{M'}/\Delta_E) \leq 0$. We estimate that the probability of the first failure case, denoted as $f_{\Delta_e=0}$, is $P(f_{\Delta_e=0}) = 0.0227$. Moreover, we find that the second failure case is a function of $\Delta_E$ which increases with model depth and, hence, the probability of failure decreases with increasing model depth. A detailed derivation can be found in Appendix B.

Therefore, we conclude that MAP inference in PCs is mildly influenced by approximate computations through AAI.

## 4.3 SAFE MULTIPLIER REPLACEMENT

Based on Eq. (9) and Eq. (10), we devise a greedy algorithm that gradually replaces multiplications with AAI while minimizing the introduced error. To do so, we heuristically select the subset of multipliers to be replaced according to $\Delta_{\mathrm{det}}$ or $\Delta_{\mathrm{dc}}$, as the error is an additive function of $\Delta_w$ terms. In particular, we compute the KL divergence for each multiplication and then greedily select those having the smallest contribution to the overall divergence. Henceforth, we can trade off approximation error and energy costs by employing heuristically selection based on $\Delta_{\mathrm{det}}$ or $\Delta_{\mathrm{dc}}$.

## 4.4 ERROR CORRECTION

The error introduced by AAI can result in substantial approximation errors in deep models as the error accumulates with an increasing number of multiplications. To reduce the error caused by AAI, [Saadat et al., 2018] proposed to correct this error by computing an *expected error*, assuming uniform probability for all possible floating-point numbers. However, in PCs this assumption will typically not hold true. Therefore, we propose to correct for the expected error w.r.t. the probability distribution represented by the circuit, *i.e.*,

$$\log \epsilon = \mathbb{E}_{\boldsymbol{x} \sim c}[\log {}^{c(\boldsymbol{x})}/_{\widetilde{c}(\boldsymbol{x})}] \tag{11}$$

and define $\log \widetilde{c}(\boldsymbol{x}) + \log \epsilon$ to be the corrected log probability of the AAI approximated circuit. Even though Eq. (11) is tractable for deterministic PCs through recursive evaluation of the circuit, it is not possible to tractably estimate the expected error for general circuits. Henceforth, we use Monte-Carlo integration to approximate the expected error, *i.e.*,

$$\log \epsilon \approx {}^{1}/_{n} \sum_{i=1}^{n} \log c(\boldsymbol{x}_i) - \log \widetilde{c}(\boldsymbol{x}_i) \tag{12}$$

with $\boldsymbol{x}_i \sim c$ drawn from the exact model. Next, we will present an algorithm summarizing our approach.

**Algorithm**  In Algorithm 1 we outline an algorithm that computes marginal queries for PCs using approximate computing with additional error correction. The algorithm iterates over all units in the circuit in topological order and computes the approximate value for each unit (line $5 - 12$) and in the end, obtains the root node value $D[c]$ (line 13). When storing the result, we correct the computation error using the pre-calculated error correction term (line 13), *c.f.* Eq. (11). For MAP queries, the sum unit is computed by applying a max operator, and error correction is not needed.

The AAI function is outlined in Algorithm 2.

## 4.5 DETAILS ON THE HARDWARE GENERATION

Computing probabilistic queries typically requires specialized hardware. For example, MAP inference requires replacing sum units with max operators and back-trace the most probable result [Choi et al., 2022, Shah et al., 2019]. We developed a specific hardware implementation for that, shown in Fig. 2(b). Every max operation has an additional OR operator ($\vee$) to record the most probable child node. Moreover, information is concatenated by AND ($\wedge$) operators at each product unit to record the most probable path. Hardware generation can then be performed by describing the PC using the hardware language Chisel, from which we can implement the PC in hardware through Verilog/VHDL.

---

**Algorithm 1** Approximate MAR Inference in PCs

**Require:** Units of $c$ in topological order $G$, test data $\mathscr{D} = \{\boldsymbol{x}_i\}_{i=1}^{n}$, error correction $\log \epsilon$
1: **function** APPROXIMATE MAR($c, G, \mathscr{D}, \log \epsilon$)
2:     $\boldsymbol{r} = \boldsymbol{0}, D = \{\}$                    ▷ Initialize variables
3:     **for all** $\boldsymbol{x}_i \in \mathscr{D}$ **do**
4:         **for all** $n \in G$ **do**
5:             **if** $n$ is sum unit $\oplus$ **then**
6:                 $D[n] \leftarrow \sum_{s_i}^{k} \text{AAI}(w_{n,s_i}, D[c_{s_i}])$
7:             **else if** $n$ is product unit $\otimes$ **then**
8:                 $D[n] \leftarrow \text{AAI}(D[c_i], D[c_j])$
9:             **else**                                        ▷ leaf
10:                 $D[n] \leftarrow p(\boldsymbol{x}_i|\theta_n)$
11:             **end if**
12:         **end for**
13:         $\boldsymbol{r}[i] \leftarrow D[c] \exp(\log \epsilon)$        ▷ Error correction
14:     **end for**
15:     **return** $\boldsymbol{r}$
16: **end function**

---

**Algorithm 2** AAI

1: **function** AAI($a, b$)
2:     $bi\_a \leftarrow a, bi\_b \leftarrow b$        ▷ represent in floating-point
3:     $bi\_r \leftarrow bi\_a + bi\_b$              ▷ binary integer addition
4:     $r \leftarrow bi\_r$                    ▷ read binary result as floating-point
5:     **return** $r$
6: **end function**

---

## 5  EXPERIMENTS

In this section, we aim to answer the following questions: ① Does AAI reduce the power consumption of multipliers on hardware? ② How does the number of bits affect the energy savings and approximation error for MAR and MAP inference in probabilistic circuits? ③ Can we reduce the energy consumption of probabilistic circuits with little or no approximation error?

**Experimental setup**  We evaluated our approach on four benchmark data sets: NLTCS, Jetser, DNA, and Book, a subset of frequently used data sets in the community (*e.g.*, [Rooshenas and Lowd, 2014]). We generated PC structures and parameters using LearnSPN [Gens and Pedro, 2013], resulting in tree-shaped non-deterministic PCs.

**Baseline comparison**  Our approach compares to quantization methods (both bit reduction and AAI quantize the signal to trade off accuracy with energy) and does not modify the PC structure (unlike pruning methods). Hence, we compare with two baselines: (1) an exact float computation with full precision (referred to as '64-bit'), standard but not always resource-efficient, and (2) a quantized exact computation, corresponding to minimum number of bits introducing the smallest error with float (referred $N_{be}$). We then compute the energy and the power consumption as listed in Table 1. The energy is the cumulative energy of every multiplier in

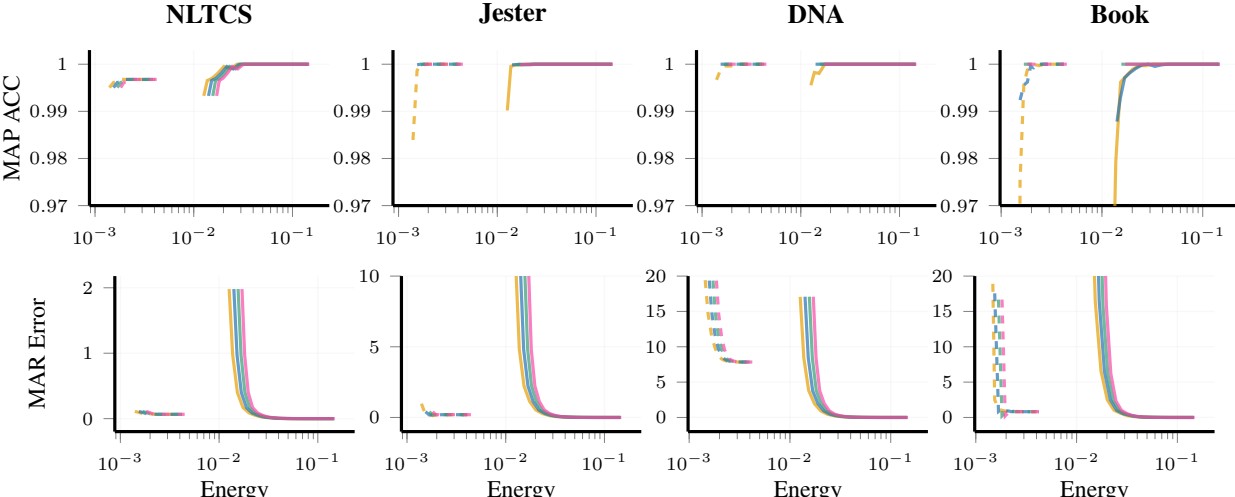

Figure 4: MAP accuracy (ACC) and MAR error for AAI (dashed) and exact (solid) multipliers for varying numbers of mantissa bits and four different numbers of exponent bits ( -- 8 bits, -- 9 bits, -- 10 bits, -- 11 bits ). All results are computed relative to exact multiplication with 64-bit. AAI significantly reduces the energy costs (x-axis) while obtaining comparable and robust results compared to exact multipliers in most cases.

the model. The power consumption of each multiplier is simulated with 65nm CMOS technology, with a unit of uW (see Appendix A.2). This value is normalized regarding the baseline energy calculated for full precision floating point.

**First hardware results on FPGA** To illustrate the memory and speed performance, we implemented one NLTCS MAR query in FPGA device Virtex7(xc7vx690) using AAI and floating point under their MAR optimal bits configuration. From Table 2, we can see that AAI uses less LUT as logic and does not use DSP blocks, AAI also consumes less memory which is realized by registers, and achieves higher maximum frequency (50 MHz for AAI and 33 MHz for floating point).

**Detailed results on simulated hardware** Hardware synthesis on FPGA requires significant time and resources, and may result in inaccurate estimates as the logic gates need to fit into existing blocks on the FPGA (i.e., FPGAs use larger computational blocks that do not fully reflect a customized hardware implementation). Hence, the more detailed experiments presented in Fig. 4, Fig. 8, Fig. 9 and Table 1 are created with an open-source Python model simulating the Chisel code with the energy model presented in Fig. 5. For each benchmark, we generated 5k samples from the exact circuit to estimate the expected error. The calculated expected error is then applied on the sample set and also on the test set to estimate the performance. Additional performance for error correction and for test set can be found in Appendix C.

**1 Does AAI reduce the power consumption?** To confirm the results of Table 2, floating-point and AAI multipliers have been designed and simulated for various resolutions

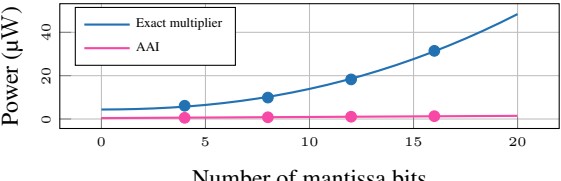

Figure 5: Power consumption of multipliers on 65nm CMOS using 8 exponent bits for increasing number of mantissa bits.

in a 65nm CMOS technology, and models have been fitted to the simulation results. Fig. 5 shows the resulting model for 8 exponent bits and varying number of mantissa bits. We see that the hardware cost is dominated by mantissa processing, and the hardware complexity grows significantly with the number of mantissa bits. As AAI uses much simpler addition hardware, the complexity and power grow linearly with the number of bits.

**2 How does the number of bits effect the energy savings and approximation errors of AAI?** In a first experiment, we replaced all multipliers with AAI to assess the error and the energy savings for MAR and MAP queries under varying resolutions. For MAP queries, we calculated the MAP inference accuracy (ACC) over the latent variables (assuming complete evidence) regarding the baseline. For MAR queries, we computed the mean log error after error correction.

Generally, as seen in Table 1, the total energy savings from 64-bit to the optimal AAI are comprised between $549\times$ and $649\times$ for MAP and between $324\times$ and $357\times$ for MAR. To evaluate the energy gains more precisely, we recorded the optimal configuration (minimum bits that keep a stable

Table 1: Overview of optimal configuration and performances over several data sets for MAP and MAR. $N_{be}$ corresponds to minimum exponent bits to ensure no underflow happens and $N_{ba}$ corresponds to the smallest error and minimum mantissa bits of exact and AAI. E represents the exponent bits, and M represents the mantissa bits. The last column indicates the energy savings obtained by AAI compared to exact multiplication in optimal configuration and exact multiplication in 64-bit.

| Query | Data set | Number of bits | | Energy | | Accuracy/Error | | Energy saving | |
| | | Exact@$N_{be}$ | AAI@$N_{ba}$ | Exact@$N_{be}$ | AAI@$N_{ba}$ | Exact@$N_{be}$ | AAI@$N_{ba}$ | $N_{be}$ | 64-bit |
|---|---|---|---|---|---|---|---|---|---|
| MAP | NLTCS | E=8, M=8 | E=8, M=5↓ | 0.02747 | 0.00182↓ | 1.00000 | 0.99677 | 15× | 549× |
| | Jester | E=9, M=6 | E=9, M=3 ↓ | 0.02165 | 0.00168↓ | 1.00000 | 1.00000 | 13× | 595× |
| | DNA | E=9, M=2 | E=9, M=2 | 0.01411 | 0.00154↓ | 1.00000 | 1.00000 | 9× | 649× |
| | Book | E=11, M=2 | E=11, M=2 | 0.01705 | 0.00182↓ | 1.00000 | 1.00000 | 9× | 549× |
| MAR | NLTCS | E=8, M=20 | E=8, M=12↓ | 0.13021 | 0.00280↓ | 0.00000 | 0.06588 | 47× | 357× |
| | Jester | E=9, M=21 | E=9, M=13↓ | 0.14521 | 0.00308↓ | 0.00002 | 0.19729 | 47× | 324× |
| | DNA | E=9, M=21 | E=9, M=13↓ | 0.14521 | 0.00308↓ | 0.00003 | 7.84791 | 47× | 324× |
| | Book | E=11, M=21 | E=11, M=9↓ | 0.14815 | 0.00280↓ | 0.00007 | 0.81912 | 53× | 357× |

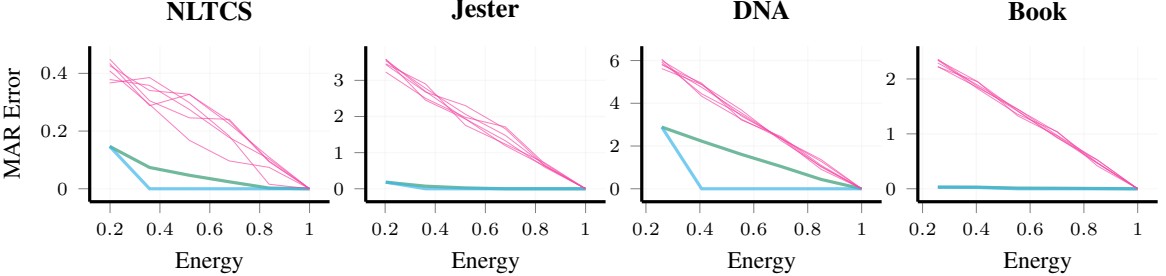

Figure 6: MAR error for partial replacement of multipliers using different strategies (— using Eq. (10), — using Eq. (9), — random). Safe selections (—, —) are superior to random selection and result in little or no error.

accuracy of the results) for both floating-point and AAI multipliers. Fig. 4 illustrates the energy savings of AAI. For MAP computation, AAI energy savings are comprised between $9\times$ and $15\times$, while maintaining a high accuracy. In addition, we did not observe an increase of the required resolution with larger model sizes. For MAR, energy savings are comprised between $47\times$ and $53\times$ with minimal error on most datasets. Moreover, we observed that AAI requires the same or fewer mantissa bits to get a stable accuracy. This is explained because floating-point requires a normalization step in Eq. (2), generating the term $M_x M_y$ or $1/2(M_x M_y)$ in the mantissa, increasing the number of mantissa bits. This term does not appear with AAI. A detailed discussion of bit requirements can be found in Appendix B. We varied the number of bits with finer granularity in Appendix C (Fig. 7).

**③ Can we reduce the energy consumption of probabilistic circuits with little or no error?** In a second experiment, we assessed how many multipliers associated with a weight/parameter can be safely replaced with approximate computing without introducing any error in the likelihood computation. For this, we used a standard 64-bit computation, where multipliers are either randomly selected (starting by replacing all multipliers associated with a weight/parameter) or incrementally replaced according to our methodology presented in Section 4.3. Note that no

Table 2: FPGA verification results on Virtex7(xc7vx690) indicating hardware block usage (less is better ↓) and achievable maximum frequency in MHz (higher is better ↑).

| | Hardware Block Usage in % | | | Max Freq. |
| | LUT ↓ | DSP ↓ | Reg(Memory) ↓ | (MHz) ↑ |
|---|---|---|---|---|
| AAI | 4.53 | 0 | 1.25 | 50 |
| Exact | 8.66 | 10 | 2.67 | 33 |

error correction is applied here. The MAR error, normalized according to the energy without any replacement, is shown in Fig. 6. To finely assess whether Eq. (10) allows for a safe multiplier selection, we also recorded the replacement ratio and the minimum relative energy without error in the Table 4 of Appendix C. Results show that we can replace a significant portion of multipliers without any error, achieving energy savings comprised between $45\%$ and $65\%$.

## 6 CONCLUSION & DISCUSSION

In this work, we introduced a dedicated approximate computing framework for energy-efficient inference in PCs on hardware. Specifically, we investigated the energy efficiency and approximation error of Addition-as-Int (AAI) multipliers in PCs for different benchmarks and query types

(evidence and MAP). We provided both a theoretical and empirical analysis of the introduced error, and our results show that maximum power savings of $649\times$ and $357\times$ can be achieved for MAP and MAR queries, respectively, while tolerating a small error. When no error is tolerated, we introduced a safe replacement method, achieving $45 - 65\%$ energy savings. Our additional error correction can reduce the approximation error for MAR queries, while MAP queries are more robust to AAI errors.

**Limitations**  This work has been demonstrated for various benchmarks and query types, proving our initial claims. A deeper analysis would be needed when implementing such methodology on other PCs structures, although we expect similar results.

**Broader Impact**  The large adoption of machine learning requires an alignment between hardware, software, and algorithms [Hooker, 2020]. Although PCs have shown great potential for solving real-world problems in reliably, their development is hampered by challenges in their hardware acceleration. This work intends to pave the way for more efficient acceleration of PCs, looking at the problem from a perspective at the interface between algorithms and hardware.

### Acknowledgements

MA acknowledges partial funding from the Research council of Fin- land through the project WHISTLE (grant number 332218). This work has also been partially funded by the European Union through the SUSTAIN project (project no. 101071179). Views and opinions expressed are, however, those of the author(s) only and do not necessarily reflect those of the European Union or EISMEA. Neither the European Union nor the granting authority can be held responsible for them. MT acknowledges funding from the Research Council of Finland (grant number 347279). We acknowledge the computational resources provided by the Aalto Science-IT project.

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

# On Hardware-efficient Inference in Probabilistic Circuits (Supplement Material)

**Lingyun Yao**[1]    **Martin Trapp**[2]    **Jelin Leslin**[1]    **Gaurav Singh**[1]    **Peng Zhang**[1]    **Karthekeyan Periasamy**[1]

**Martin Andraud**[1,3]

[1]Department of Electronics and Nanoengineering, Aalto University, Espoo, Finland
[2]Department of Computer Science, Aalto University, Espoo, Finland
[3]ICTEAM, UCLouvain, Louvain-La-Neuve, Belgium

## A    TECHNICAL DETAILS

### A.1    DETAILS ABOUT EXACT FLOATING-POINT MULTIPLICATIONS

Multiplying two floating-point numbers $x$ and $y$, with respective exponent bits $E_x, E_y$ and mantissa bits $M_x, M_y$ is done in four steps: (1) Exponent addition $E_x + E_y$, (2) mantissa multiplication $(1 + M_x) \times (1 + M_y)$, (3) Exponent normalization to ensure mantissa value between 0 and 1, and (4) Rounding mantissa value into limited mantissa bits. The product $x \cdot y$ is:

$$x \cdot y = 2^{E_x + E_y}(1 + M_x)(1 + M_y) \tag{13}$$

$$= \begin{cases} 2^{E_x + E_y}(1 + M_x + M_y + M_x M_y) & \text{if } M_x + M_y + M_x M_y < 1 \\ 2^{E_x + E_y + 1}(1 + \tfrac{1}{2}(M_x + M_y + M_x M_y - 1)) & \text{otherwise} \end{cases} \tag{14}$$

A few observations should be made. First, after normalization (step (3) above), the original mantissa value can exceed number 1 which is the maximum mantissa value the mantissa bits can represent. This is taken care of by increasing the exponent (*i.e.* multiply by two) and dividing the overall mantissa result by two (see Eq. (14)). Second, mantissa rounding (step (4) above) can lead to small errors due to the finite precision. Third, a bias can be added to the exponent value, shifting the range that exponent bits can represent. In the example of PCs, as exponent values are smaller than 0, a negative bias can be added to the exponent so that the exponent is always coded with positive integers. Adding a bias term does not influence the analysis, thus it is omitted for clarity.

### A.2    DETAILS ABOUT HARDWARE GENERATION

#### A.2.1    Theoretical analysis of computation resolution

The theoretical analysis provides a starting point for the resolution analysis, ensuring that the PC can be represented with minimal error on *e.g.* 32 or 64 bits. It primarily aims to: 1.) avoid underflow, as the PC processes small numbers, and 2.) ensure that the representation error of any value is smaller than a pre-defined error setting. To avoid underflow, the analysis determines the smallest positive non-zero probability at the top node, denoted as $MV$. $MV$ is found by replacing all sum nodes by *min* operators (i.e., $\text{Out} = \min(\text{In}_1, \text{In}_2)$ ) and traversing the graph bottom-up, with all leaf node indicators set to '1' [Shah et al., 2019]. In terms of representation error, the analysis uses relative error tolerance as a constraint, given by $\epsilon = |\tilde{x} - x|/|x|$, where and $x$ is the real value of the number and $\tilde{x}$ its represented value. $\epsilon$ is then compared with the quantization error of each representation. The final resolution requirements can be determined with $MV$ and $\epsilon$. We explicit the analysis with a fixed-point representation first and expand to float.

**Fixed-point Analysis (FxP)**. Assume a $N$-bits fixed-point representation with $I$ integer bits and $F$ fraction bits. As the PC computes probabilities, the top node value cannot exceed 1; hence $I = 1$ is chosen. $F$ depends on the minimum value that

*Accepted for the 40th Conference on Uncertainty in Artificial Intelligence* (UAI 2024).

Table 3: Energy from 65nm CMOS technology.

| Operators | Power@E,M(uW) | | | |
|---|---|---|---|---|
| | 8,4 | 8,8 | 8,12 | 8,16 |
| floating-point | 6.121 | 9.889 | 18.27 | 31.45 |
| AAI | 0.55384 | 0.79835 | 1.0482 | 1.2996 |

can be encoded, $1/2^F$. The fixed-point quantization $\text{QE}_{\text{FxP}}$ can be written as:

$$\text{QE}_{\text{FxP}} = \frac{1}{2}\frac{1}{2^F} = 2^{-(F+1)} \tag{15}$$

As explained earlier $\text{QE}_{\text{FxP}}$ should be in the error bound for $\tilde{\cdot}$ fixed by $MV$ and $\epsilon$. In the worst case, it corresponds to $\text{QE}_{\text{FxP}} = MV(1+\epsilon) - MV(1-\epsilon) = 2MV\epsilon$. Thus, the minimum required number of fraction bits $F_{\min}$ to be used is given as:

$$F_{\min} = \lceil \log_2(\frac{1}{2MV\epsilon}) \rceil \tag{16}$$

**Floating-point Analysis (FlP)** Assume an FlP representation with $E$ exponent bits and $M$ mantissa bits. In essence, exponent bits $E$ represent the binary point of the number (i.e., the range), and mantissa bits $M$ further quantize values within this range. First, the smallest encoded exponent value should cover $MV$. The encoded exponent value $2^{E_{min}}$ can be determined based on the fixed point analysis, as $MV$ is the same (see eq. 16), i.e., $2^{E_{\min}} \geq F_{\min}$. This gives:

$$E_{\min} = \lceil \log_2(F_{\min}) \rceil \tag{17}$$

Where $\lceil \rceil$ represents rounding up to the next integer. Second, $M$ is adjusted to satisfy the precision based on the error tolerance $\epsilon$. Specifically, $M$ should be one order lesser than the target error tolerance $\epsilon$. For example, if $\epsilon = 10^{-P}$, a minimum of $P+1$ digits of precision is needed, giving $2^M > 10^{P+1}$. Only one integer bit is considered (as in the FxP analysis), which adds an extra mantissa bit. The final mantissa bits $M_{\text{req}}$ is given by:

$$M_{\text{req}} = \lceil \log_2(10^{P+1}) \rceil \tag{18}$$

### A.2.2 Energy model

Floating-point and AAI multipliers have been designed and simulated for various resolutions in a 65nm CMOS technology. Specifically, we synthesized the design using 65nm high threshold voltage (HVT) standard cells for various resolution bits and estimated the power from the synthesized netlist.

For these measurements, we performed model-fitting of the floating-point multiplier with the function $k_m(M+1)^2 \log(M+1) + k_e E$. Function attained in [Shah et al., 2019] is fitted with the function $k_m(M+1)^2 \log(M+1)$, however, according to Appendix A, classic floating-point multiplication also contains exponent addition which contributes limited energy consumption that ignored by [Shah et al., 2019]. Here we add this part into consideration and it fits well with our 65nm CMOS technology results. We fit the model of the AAI multiplier with the function $k_a(M+E)$. As the energy linear grows with the whole bits number, it also fits well with our 65nm CMOS technology results. Energy raw data can be found in Table 3.

The final energy model for the floating-point multiplier and AAI multiplier are $0.0328(M+1)^2 \log(M+1) + 0.5469E$ and $0.0520160465095606(M+E)$ respectively. At 32 bits, AAI uses 1.664μW compared to 64.413μW for the exact multiplier, saving 38x. At 64 bits, AAI uses 3.329μW compared to 371.791μW for the exact multiplier, which saves 112×.

# B DERIVATIONS

## B.1 THEORETICAL ERROR ANALYSIS

Given a PC $c$ and the same PC but using approximate multipliers $\widetilde{c}$ we may compute the KL divergence between both, given by

$$D_{KL}\left(c||\widetilde{c}\right) = \int c(\boldsymbol{x})\left(\log c(\boldsymbol{x}) - \log \widetilde{c}(\boldsymbol{x})\right) d\boldsymbol{x}. \tag{19}$$

Note that we are assuming $c(\star) = 1$ without loss of generality.

In the following, we will utilize the shallow mixture representation of a PC to simplify the derivations. For this, we briefly recall relevant concepts and refer to [Trapp et al., 2019] for detailed derivations.

**Definition B.1** (Induced tree [Zhao et al., 2016]). *Let an $c$ with computational graph $\mathcal{G}$, parametrized by $\boldsymbol{w}, \boldsymbol{\theta}$, and scope function $\psi(\cdot)$ be given.*

*Consider a sub-graph $\mathcal{T}$ of $\mathcal{G}$ obtained as follows: i) for each sum unit, delete all but one outgoing edge and ii) delete all nodes and edges which are now unreachable from the root. Any such tree $\mathcal{T}$ is called an* induced tree.

Note that a PC can always be written as the mixture of induced trees, *i.e.*,

$$c(\boldsymbol{x}) = \sum_{t=1}^{\tau} \underbrace{\prod_{w \in \mathcal{T}_t} w}_{=p(\mathcal{T}_t)} \prod_{L \in \mathcal{T}_t} L(\boldsymbol{x}_L), \tag{20}$$

where the sum runs over all possible induced trees in $c$ ($\tau$ many), and $L(\boldsymbol{x}_L)$ denotes the evaluation of leaf unit $L$ on the restriction of $\boldsymbol{x}$ to $\psi(L)$.

In case $c$ is deterministic, then every $\boldsymbol{x}$ is associated with a single unique induced tree, *i.e.*, the probability of all other induced trees is zero, by definition of determinism and we use $\mathcal{T}(\boldsymbol{x})$ to denote the associated induced tree.

### B.1.1 Deterministic PCs

If $c$ (and consequently also $\widetilde{c}$) is deterministic, then we can compute the Eq. (19) exactly, *i.e.*,

$$D_{KL}\left(c||\widetilde{c}\right) = \int c(\boldsymbol{x})\left(\log c(\boldsymbol{x}) - \log \widetilde{c}(\boldsymbol{x})\right) d\boldsymbol{x} = \int c(\boldsymbol{x}) \sum_{w_i \in \mathcal{T}(\boldsymbol{x})} \Delta_{w_i} d\boldsymbol{x}, \tag{21}$$

$$= \sum_{t=1}^{\tau} p(\mathcal{T}_t) \sum_{w_i \in \mathcal{T}} \Delta_{w_i} = \sum_{t=1}^{\tau} \sum_{w_i \in \mathcal{T}} p(\mathcal{T}_t)\Delta_{w_i} \tag{22}$$

$$= \sum_{w_i \in c} \sum_{\substack{\mathcal{T} \in c \\ w_i \in \mathcal{T}}} p(\mathcal{T})\Delta_{w_i} = \sum_{w_i \in c} \Delta_{w_i} \underbrace{\sum_{\substack{\mathcal{T} \in c \\ w_i \in \mathcal{T}}} p(\mathcal{T})}_{\leq 1 \text{ by assumption}} = \Delta_{dc}, \tag{23}$$

where $\sum_{\substack{\mathcal{T} \in c \\ w_i \in \mathcal{T}}} p(\mathcal{T}_t)$ can be efficiently computed through a bottom-up path by setting all indicators below $w_i$ to one and all other indicators to zero.

### B.1.2 Non-deterministic PCs

We will now investigate the case of non-deterministic PCs, for which it is well-known that the KL divergence can no be evaluated analytically. First, recall the objective, *i.e.*,

$$D_{KL}(c||\widetilde{c}) = \int c(\boldsymbol{x})\left(\log c(\boldsymbol{x}) - \log \widetilde{c}(\boldsymbol{x})\right) d\boldsymbol{x} \tag{24}$$

$$= \underbrace{\int c(\boldsymbol{x})\left(\log c(\boldsymbol{x}) - \log \widetilde{c}(\boldsymbol{x})\right) d\boldsymbol{x}}_{=\Delta_{dc}}, \tag{25}$$

where we will focus on approximating $\Delta_{dc}$. For this, let us rewrite $\Delta_{dc}$ by expanding the density function of $c$ in form of a shallow mixture, *i.e.*,

$$\Delta_{dc} = \int c(\boldsymbol{x})\left(\log \frac{c(\boldsymbol{x})}{\widetilde{c}(\boldsymbol{x})}\right) d\boldsymbol{x} \tag{26}$$

$$= \int c(\boldsymbol{x})\left(\log\left[\sum_{t=1}^{\tau} p(\mathcal{T}_t)\prod_{\mathsf{L}\in\mathcal{T}_t} p(\boldsymbol{x}\mid\theta_{\mathsf{L}})^{1/\widetilde{c}(\boldsymbol{x})}\right]\right) d\boldsymbol{x} \tag{27}$$

$$= \int c(\boldsymbol{x})\left(\log \sum_{t=1}^{\tau} \exp\left[\log p(\mathcal{T}_t) + \sum_{\mathsf{L}\in\mathcal{T}_t}\log p(\boldsymbol{x}\mid\theta_{\mathsf{L}}) - \log \widetilde{c}(\boldsymbol{x})\right]\right) d\boldsymbol{x}, \tag{28}$$

where we note that $\log \widetilde{c}(\boldsymbol{x})$ simplifies under discrete data, *i.e.*,

$$\log \widetilde{c}(\boldsymbol{x}) = \log\left(\sum_{t=1}^{\tau}\widetilde{p}(\mathcal{T}_j)\underbrace{\widetilde{p}(\boldsymbol{x}\mid\mathcal{T}_t)}_{=\mathbb{1}[\boldsymbol{x}\in\mathcal{T}_t]}\right) \qquad\text{(assuming discrete data)}$$

$$= \log\left(\sum_{\substack{t=1\\\boldsymbol{x}\in\mathcal{T}_t}}^{\tau}\widetilde{p}(\mathcal{T}_t)\right), \tag{29}$$

as, all leaves are indicator functions, simplifying the expression for $\log \widetilde{c}(\boldsymbol{x})$ as the log over the sum of all induced trees for which all indicator leaves return one.

First note that $\log(x+y) = \log(x) + \log(1 + \frac{y}{x}) \approx \log(x) + \frac{y}{x}$, with least error if $y \leq x$. Secondly, given a permutation $\sigma_{\boldsymbol{x}}(\cdot)$ of the induced trees in the circuit such that $\widetilde{p}(\mathcal{T}_{\sigma_{\boldsymbol{x}}(1)}) \geq \widetilde{p}(\mathcal{T}_{\sigma_{\boldsymbol{x}}(2)}) \geq \cdots > \widetilde{p}(\mathcal{T}_{\sigma_{\boldsymbol{x}}(k)}) \geq \widetilde{p}(\mathcal{T}_{\sigma_{\boldsymbol{x}}(k+1)}) = \cdots = \widetilde{p}(\mathcal{T}_{\sigma_{\boldsymbol{x}}(\tau)}) = 0$, we can approximate Eq. (29) as follows:

$$\log \widetilde{c}(\boldsymbol{x}) = \log \widetilde{p}(\mathcal{T}_{\sigma_{\boldsymbol{x}}(1)}) + \log\left(1 + \frac{\sum_{j=2}^{k}\widetilde{p}(\mathcal{T}_{\sigma_{\boldsymbol{x}}(2)})}{\widetilde{p}(\mathcal{T}_{\sigma_{\boldsymbol{x}}(1)})}\right) \tag{30}$$

$$\approx \log \widetilde{p}(\mathcal{T}_{\sigma_{\boldsymbol{x}}(1)}) + \sum_{j=2}^{k}\widetilde{p}(\mathcal{T}_{\sigma_{\boldsymbol{x}}(j)}). \qquad\text{(Taylor approx.)}$$

Consequently, we get that

$$\Delta_{dc} \approx \int c(\boldsymbol{x})\left(\log \sum_{t=1}^{k}\exp\left[\log p(\mathcal{T}_{\sigma_{\boldsymbol{x}}(t)}) - \left(\log \widetilde{p}(\mathcal{T}_{\sigma_{\boldsymbol{x}}(1)}) + \sum_{j=2}^{k}\widetilde{p}(\mathcal{T}_{\sigma_{\boldsymbol{x}}(j)})\right)\right]\right) d\boldsymbol{x} \qquad\text{(assuming discrete data)}$$

$$= \int c(\boldsymbol{x})\left(\sum_{w\in\mathcal{T}_{\sigma_{\boldsymbol{x}}(t)}}\Delta_w - \sum_{j}^{k}\widetilde{p}(\mathcal{T}_{\sigma_{\boldsymbol{x}}(j)})\right) d\boldsymbol{x} + \int c(\boldsymbol{x})\log\left(1 + \sum_{t=2}^{k}\frac{p(\mathcal{T}_{\sigma_{\boldsymbol{x}}(t)})}{p(\mathcal{T}_{\sigma_{\boldsymbol{x}}(1)})}\right) d\boldsymbol{x} \tag{31}$$

$$= \int c(\boldsymbol{x})\left(\sum_{w\in\mathcal{T}_{\sigma_{\boldsymbol{x}}(t)}}\Delta_w - \sum_{j}^{k}\widetilde{p}(\mathcal{T}_{\sigma_{\boldsymbol{x}}(j)})\right) d\boldsymbol{x} + C. \tag{32}$$

Unfortunately, the above approximation is still not analytically tractable, but can be approximated using Monte-Carlo integration.

## B.2 MAP ERROR

When performing MAP queries, we are interested in estimating the sensitivity of MAP inference to approximate computing. For this, we examine whether the most probable path in the circuit stays the same when replacing exact multipliers by AAI.

First, recall that in case of MAP inference, sum units are replaced by max operations [Choi, 2022]. Our analysis is based on a max operation over two sub-circuits, denoted as left ($l$) and right ($r$) branches respectively. Each branch/sub-circuit is a binary product over inputs denoted as $x_l, x_r$ and $y_l, y_r$ respectively. Let, $M_{x_l}, M_{y_l}$ denote the input's mantissa values in the left branch, and $M_{x_r}, M_{y_r}$ in the right, and recall that all mantissa values are $M \in [0, 1]$. We will refer to the computation path using exact floating-point multipliers as the *exact* path, and the path using AAI multipliers as the *approximate* path. In the exact path, assuming the left branch is the most probable, we can write:

$$2^{E_{x_l}+E_{y_l}}(1 + M_{x_l})(1 + M_{y_l}) > 2^{E_{x_r}+E_{y_r}}(1 + M_{x_r})(1 + M_{y_r}) \quad (33)$$

applying the log transform we obtain

$$E_{x_l} + E_{y_l} + \log_2(1 + M_{x_l}) + \log_2(1 + M_{y_l}) > E_{x_r} + E_{y_r} + \log_2(1 + M_{x_r}) + \log_2(1 + M_{y_r}) \quad (34)$$

$$\underbrace{E_{x_l} + E_{y_l} - (E_{x_r} + E_{y_r})}_{\Delta_E} + \underbrace{\log_2(1 + M_{x_l}) + \log_2(1 + M_{y_l}) - (\log_2(1 + M_{x_r}) + \log_2(1 + M_{y_r}))}_{\Delta_M} > 0. \quad (35)$$

In the approximate path, the same analysis gives:

$$E_{x_l} + E_{y_l} + M_{x_l} + M_{y_l} > E_{x_r} + E_{y_r} + M_{x_r} + M_{y_r} \quad (36)$$

$$\Delta_E + \underbrace{M_{x_l} + M_{y_l} - (M_{x_r} + M_{y_r})}_{\Delta_{M'}} > 0. \quad (37)$$

We will now consider the condition for which the most probable path under AAI is not the same as in case of exact computations. We start with the two following expressions: (1): $(\Delta_E + \Delta_M) > 0$, representing the exact path, and (2): $(\Delta_E + \Delta_{M'}) > 0$, representing the approximate path. And look at the product between the exact and approximate paths: (3): $(\Delta_E + \Delta_M)(\Delta_E + \Delta_{M'})$. If expressions (1) and (2) are both true or both false, then their product (3) is always positive. Hence, both paths indicate the same most probable branch. Any discrepancy between both paths happens when either (1) or (2) is true, and the other is false. Consequently, their product becomes negative or zero in this case, which we refer to as the *failure* condition $f$, *i.e.*,

$$f \rightarrow (\Delta_E + \Delta_M)(\Delta_E + \Delta_{M'}) \leq 0 \quad (38)$$

if $\Delta_E = 0$ then:

$$f_{\Delta_E=0} \rightarrow (\Delta_M)(\Delta_{M'}) \leq 0 \quad (39)$$

and if $\Delta_E \geq 1$

$$f_{\Delta_E \geq 1} \rightarrow (1 + \Delta_M/\Delta_E)(1 + \Delta_{M'}/\Delta_E) \leq 0. \quad (40)$$

Note that $\Delta_E$ is a difference function of integers and, hence, can only take integer values. Moreover, if $\Delta_E \leq -1$ we can flip the inequality in Eq. (35) and Eq. (35) while still testing for consistency of both computations. Hence, the argument above does not change.

The probability of failure $P(f_{\Delta_e=0})$ for $f_{\Delta_e=0}$ is given as:

$$P(f_{\Delta_e=0}) = \int_0^1 \int_0^1 \int_0^1 \int_0^1 f_{\Delta_E=0} \, \mathrm{d}M_{x_l} \mathrm{d}M_{y_l} \mathrm{d}M_{x_r} \mathrm{d}M_{y_r} \approx 0.0227 \quad (41)$$

which we estimated using Monte Carlo integration under the assumption that the Mantissa values are uniformly distributed.

The second failure case, Eq. (40), is a function of $\Delta_E$. Note that $\Delta_E$ increases as a function of the number of multiplications, hence, restricting the probability of failure in PCs for larger models.

## B.3 INFLUENCE OF THE NUMBER OF BITS

Let us first considering a floating-point multiplier computing the product of $x$ and $y$, *i.e.*,

$$2^{E_x+E_y}(1+M_x)(1+M_y) = 2^{(E_x+E_x+E_f)}(1+M_f), \tag{42}$$

where $E_f \geq 0$, $E_x \leq 0$, $E_y \leq 0$, and $0 < M_f < 1$ denote the new mantissa value $M_f$ and an exponent carry. Let use examine the computation of the exponent carry and the new mantissa value. If $(M_x + M_y + M_xM_y) < 1$ then:

$$E_f = 0, \qquad M_f = M_x + M_y + M_xM_y \tag{43}$$

otherwise:

$$E_f = 1, \qquad M_f = \frac{M_x + M_y + M_xM_y - 1}{2}. \tag{44}$$

Note that in floating-point representation, mantissa bits depend on the smallest value. First, assume we need $N$ bits for $M_x$ and $M_y$, in the worst case we need $2N$ bits for $M_xM_y$ and, therefore, $2N$ bits for $M_x + M_y + M_xM_y$ in the worst case for the case that $(M_x + M_y + M_xM_y) < 1$. In case $(M_x + M_y + M_xM_y) \geq 1$, we need to right shift $M_x + M_y + M_xM_y$ by one one bit to ensure the mantissa value stays within the range $[0, 1]$ and increase the exponent. This leads to one additional bit for the mantissa value, and in the worst case, we need $2N + 1$ bits for $\frac{M_x+M_y+M_xM_y-1}{2}$.

Now let us consider an AAI multiplier computing the product of $x$ and $y$, *i.e.*,

$$E_x + E_y + M_x + M_y = E_x + E_y + E_a + M_a, \tag{45}$$

where $E_a \geq 0$, $E_x \leq 0$, $E_y \leq 0$, and $0 < M_a < 1$

Again, we can examine the computation of $E_a$ and $M_a$. If $(M_x + M_y) < 1$ then:

$$E_a = 0, \qquad M_a = M_x + M_y \tag{46}$$

otherwise:

$$E_a = 1, \qquad M_a = M_x + M_y - 1 \tag{47}$$

Again, the number of required mantissa bits still depends on the smallest value. In case $(M_x + M_y) < 1$, there is no carry value so the mantissa bits requirements stay the same. In the second case, *i.e.*, $(M_x + M_y + M_xM_y) \geq 1$, the carry '1' goes to the exponent bits and the mantissa bits requirements still stay the same.

Therefore, we can conclude that AAI computation requires similar number of exponent bits but fewer mantissa bits.

# C  EXPERIMENTAL DETAILS

## C.1  INFRASTRUCTURE DETAILS

We ran all experiments with the Python model on a SLURM cluster. We parallelly computed for each configuration and collected the data for analysis.

## C.2  EXPERIMENTS RESULTS

We first calculated the expected error and applied this to the sample set by 4 exponent bits(8, 9, 10, 11) and mantissa bits(from 2 to 21). Fig. 7 comes from the same data as Fig. 4, but Fig. 7 shows the accuracy or error by varying numbers of exponent and mantissa bits, while Fig. 4 shows the accuracy or error by energy.

Fig. 8 shows the performance of error correction on the sample, the top dashed line is the result before error correction, and the bottom dashed line is the result after error correction. Fig. 9 shows the performance of error correction on the test set, the top dashed line is the result before error correction, and the bottom dashed line is the result after error correction. We can see the performance on the test set is similar, the error correction significantly reduced the error on the test set.

Table 4: Minimum Normalized Energy without error under Replacement.

| Data set | Replacement ratio | Normalized Energy | Error |
|---|---|---|---|
| NLTCS | 0.646 | 0.35836 | 0.00000 |
| Jester | 0.643 | 0.36140 | 0.00000 |
| DNA | 0.598 | 0.40644 | 0.00000 |
| Book | 0.448 | 0.55298 | 0.00000 |

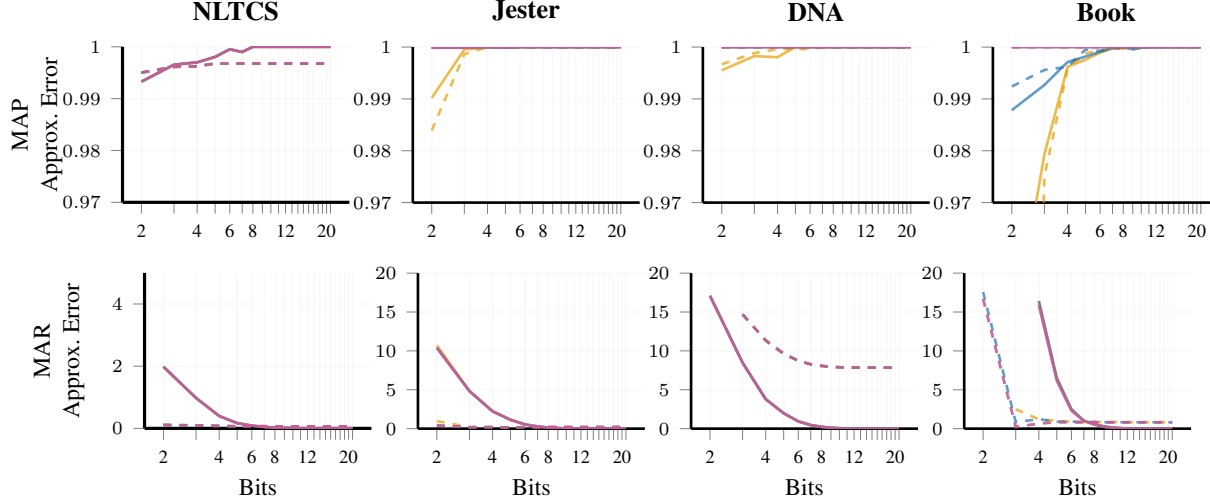

Figure 7: Results for AAI (dashed lines) and exact (solid lines) multipliers using varying numbers of exponent and mantissa bits ( -- 8 exponent bits, -- 9 exponent bits, -- 10 exponent bits, -- 11 exponent bits ).

For safe replacement, we started by replacing all multipliers associated with a weight/parameter(not all multipliers) and ended with no replacement. In our experiments, we count how much percentage we replaced in the bottom layer(multipliers associated with a weight/parameter) from 0 to 100 percent, with a step of 20 percent. We recorded the maximum replacement ratio of the whole multipliers in our experiments and the corresponding normalized energy which keeps the 0 error in Table 4.

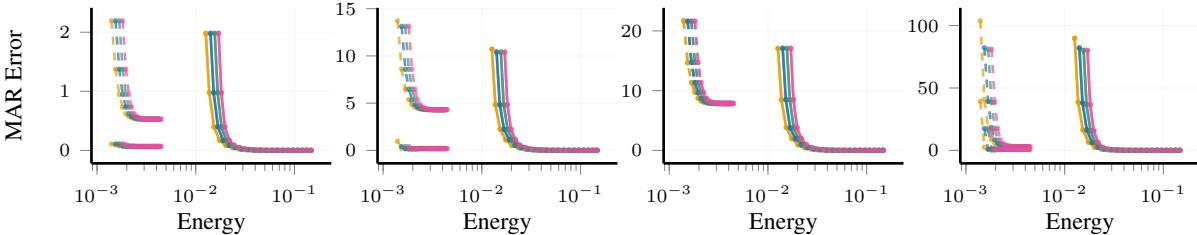

Figure 8: Results for AAI (dashed lines) and exact (solid lines) multipliers using varying the number of exponent and mantissa bits on the sample set with error correction. The top dashed line is the result before error correction, and the bottom dashed line is the result after error correction ( -- 8 exponent bits, -- 9 exponent bits, -- 10 exponent bits, -- 11 exponent bits ).

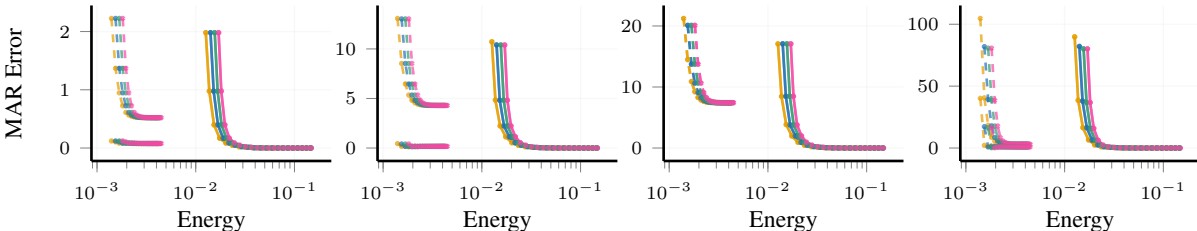

Figure 9: Results for AAI (dashed lines) and exact (solid lines) multipliers using varying the number of exponent and mantissa bits on the test set with error correction. The top dashed line is the result before error correction, and the bottom dashed line is the result after error correction. ( -- 8 exponent bits, -- 9 exponent bits, -- 10 exponent bits, -- 11 exponent bits ).

