# OpenReview forum: "On Hardware-efficient Inference in Probabilistic Circuits"
_auai.org/UAI/2024/Conference — UAI 2024 poster_

### Official Review · Reviewer_sik5 · 2024-03-04

**Q2-1 Originality-Novelty:** 3
**Q2-2 Correctness-Technical Quality:** 3
**Q2-5 Clarity Of Writing:** 4

**Q1 Summary And Contributions:**

The paper presents a possible way to replace safe multipliers with Addition-as-Int gates to cope with probabilistic circuit. The paper also discusses about the accuracy loss and addresses the problem by proposing a way to reduce the loss.
The proposal have been tested on 4 benchmarks, showing good results, both in terms of efficiency and accuracy loss.

**Q2-3 Extent To Which Claims Are Supported By Evidence:**

3: Good: the main claims are supported by convincing evidence (in the form of adequate experimental evaluation, proofs, (pseudo-)code, references, assumptions).

**Q2-4 Reproducibility:**

3: Good: key resources (e.g. proofs, code, data) are available and key details (e.g. proofs, experimental setup) are sufficiently well-described for competent researchers to confidently reproduce the main results.

**Q3 Main Strengths:**

The results are very promising. The proposal seems ready to be widely applied.

**Q4 Main Weakness:**

The tests are too few to ensure the generality of the proposal.

**Q5 Detailed Comments To The Authors:**

I have no particular comments to make. I found the document clear and solid. The results are very promising and I did not find any flaws.

I would suggest re-reading the paper because I found several errors in English.

**Q9 Complying With Reviewing Instructions:**

Yes

---

> ### Author Rebuttal · Authors · 2024-04-04
>
> Thank you for your recognition of our work!
>
> Answer to Q4:
> We agree that additional experiments, especially in real-world settings, would be interesting. Given that our paper is a proof of concept, we will leave this for future work.
>
> Answer to Q5:
> Thanks for the careful reading, we will re-read the manuscript and correct typos.

---

### Official Review · Reviewer_HRxq · 2024-03-18

**Q2-1 Originality-Novelty:** 2
**Q2-2 Correctness-Technical Quality:** 3
**Q2-5 Clarity Of Writing:** 3

**Q10 Ethical Concerns:**

-

**Q1 Summary And Contributions:**

This is an exciting study in the field of probabilistic circuits. The authors consider the potential trade-off between energy saving and errors in the presence of an approximation (based on Mitchell's method) for the local floating-point operations performed by hardware inference in a probabilistic circuit and related to the fact that the operations are performed in the log domain to prevent underflows.

**Q2-3 Extent To Which Claims Are Supported By Evidence:**

3: Good: the main claims are supported by convincing evidence (in the form of adequate experimental evaluation, proofs, (pseudo-)code, references, assumptions).

**Q2-4 Reproducibility:**

4: Excellent: key resources (e.g. proofs, code, data) are available and key details (e.g. proof sketches, experimental setup) are comprehensively described for competent researchers to confidently and easily reproduce the main results.

**Q3 Main Strengths:**

The problem of optimising hardware-based inference is relatively new and unexplored, especially in the ProcCirc area.
The potential for savings is considerable (as the model sizes reported in Fig1 are realistic for real-world applications). The experiments confirm this point.

**Q4 Main Weakness:**

As the authors noticed in the last section, no significant examples of real hardware-based approaches to PC inference exist. This makes the practical relevance of this paper low, but the situation might quickly change if the adoption of PCs by the ML community increases.
The authors focus on four datasets of mid/large sizes from a classical repository. Adopting larger datasets leading to larger circuits (ex., from the same repository) would make the empirical results stronger.

**Q5 Detailed Comments To The Authors:**

Why not consider some larger datasets from the Lowd & Davis repository?
I have found the theoretical analysis of the MAP queries unclear. In particular, I think the integral in (41) is based on some "uniformity" assumption that might not be tenable in many cases.
Figure 3 is not very clear to me.

**Q9 Complying With Reviewing Instructions:**

Yes

---

> ### Author Rebuttal · Authors · 2024-04-04
>
> Thank you for your thorough analysis and all the constructive feedback!
>
> Regarding Q4:
> The goal of our paper is a proof of concept, showing that the hardware energy costs of PCs can be substantially reduced with relatively simple custom hardware. Even though evaluating our method on even larger data sets would be interesting, we believe that our current experimental setup sufficiently supports our claims. Note that computations on hardware get increasingly more challenging the larger the circuit is, as the requirement on the number of bits required for accurate computations increases drastically. We believe that more research is needed for the acceleration of large-scale models and hope that our contribution encourages more research to work on the hardware acceleration of circuit models.
> Note that we are currently planning to design custom hardware based on our proposal, but this is out of the scope of this work.
>
> Regarding the theoretical analysis of the MAP queries  and Pequation(41):
> The reviewer is correct that Eq. (41) is based on the assumption that Mantissa values are uniformly distributed. We agree it will not exactly be uniformly distributed in all cases. Still, we did the integration in different types of distributions, and the result stayed similar and low. We will clarify this in the final version.
>
> Regarding Figure 3:
> This figure plots the value before using Mitchell’s approximation: f(F)=log2(1+F) and after Mitchell’s approximation: f(F)=F, to compare the error in different mantissa values F(X-axis). The error is due to the difference of log2(1+F) and F.
> We can see the error is 0 when the mantissa is 0 or 1 and Mitchell’s approximation underestimates the true value in between. We will explain it more clearly in our manuscript.

---

### Official Review · Reviewer_yoZE · 2024-03-20

**Q2-1 Originality-Novelty:** 3
**Q2-2 Correctness-Technical Quality:** 3
**Q2-5 Clarity Of Writing:** 3

**Q10 Ethical Concerns:**

No.

**Q1 Summary And Contributions:**

The paper introduces an approach for efficient inference on probabilistic circuits that replaces some (or all) multiplication operations with a more hardware-efficient (slightly approximate) alternative. Using a custom error-correction mechanism, the presented approach can reduce the energy consumption of different types of probabilistic inference queries several hundred times with minimal accuracy loss. It is also shown how one can greedily replace about 45%-65% of multiplication operations without incurring any accuracy loss and achieving 45%-65% energy savings.

**Q2-3 Extent To Which Claims Are Supported By Evidence:**

4: Excellent: all claims are supported by very convincing evidence (in the form of comprehensive experimental evaluation, rigorous mathematical proofs, detailed (pseudo-)code, precise references, well-motivated and realistic assumptions) and the authors deliver what they promise.

**Q2-4 Reproducibility:**

4: Excellent: key resources (e.g. proofs, code, data) are available and key details (e.g. proof sketches, experimental setup) are comprehensively described for competent researchers to confidently and easily reproduce the main results.

**Q3 Main Strengths:**

The paper is technically sound, convincing, and contains all the details required for reproducibility. Although building on several recent prior works, the paper adapts them to the setting of probabilistic circuits in a non-trivial way and with extensive theoretical and experimental analysis.

**Q4 Main Weakness:**

The presentation of the paper could be improved. Specifically by:
1) clarifying various bits of unintroduced notation (e.g., \widetilde{p}(\mathcal{T}_{\sigma_{\mathbf{x}}(j)})),
2) explaining what weights, subtrees, etc. the sums/integrals are over (and why), particularly in Section 4.1,
3) highlighting the theoretical contribution by formulating them as theorems.

**Q5 Detailed Comments To The Authors:**

* Although this is done to some extent, the paper would benefit from more forward references and linking statements that clarify what the reader should expect from each section and subsection. For example:
1) end the introductory paragraph of Section 4 with an overview of all of the subsections, and
2) when referring to AAI early in the paper, mention where it is being defined/described.
* Can you support generic statements such as "Computing probabilistic queries typically requires specialized hardware" and "their development is hampered by challenges in their hardware acceleration" with references?
* Some typos:
** "additions ( using" -> "additions (using"
** "exiting" -> "existing"
** "Smooth &" -> "Smoothness &"
** "inetger" -> "integer"
** "in turns" -> "in turn"
** "Eq. (8)" -> "Eq. (7)"
** "approximation on the KL" -> "approximation of the KL divergence"
** "are the difference" -> "are the differences"
** "x" -> "x_i" (on line 10 of Algorithm 1)
** "we are aim" -> "we aim"
** "effect" -> "affect"
** "referred (as)" -> "referred to as"
** "Mhz" -> "MHz"
** "Domingos Pedro" -> "Pedro Domingos"

**Q9 Complying With Reviewing Instructions:**

Yes

---

> ### Author Rebuttal · Authors · 2024-04-04
>
> Thanks for the valuable feedback and all the detailed comments!
>
> Regarding Q4
> Thanks for pointing out the detailed directions to improve the presentation of the paper. We will clarify the notation and concepts in our paper accordingly and highlight the main theoretical contributions by formulating them as theorems. Moreover, we will use the extra space to add an additional introductory paragraph where needed.

---

### Official Review · Reviewer_PtbY · 2024-03-21

**Q2-1 Originality-Novelty:** 2
**Q2-2 Correctness-Technical Quality:** 3
**Q2-5 Clarity Of Writing:** 3

**Q1 Summary And Contributions:**

Hardware acceleration of probabilistic circuits poses several challenges, in particular the irregularity of the graphs (reducing parallelism) and the necessary resolution of computations. This article provides solutions for the latter. Using Mitchell's approximation and Addition as Init techniques the authors introduce a novel approximation for multiplications in hardware and demonstrate their results using some experiments and FPGA implementation.

**Q2-3 Extent To Which Claims Are Supported By Evidence:**

3: Good: the main claims are supported by convincing evidence (in the form of adequate experimental evaluation, proofs, (pseudo-)code, references, assumptions).

**Q2-4 Reproducibility:**

3: Good: key resources (e.g. proofs, code, data) are available and key details (e.g. proofs, experimental setup) are sufficiently well-described for competent researchers to confidently reproduce the main results.

**Q3 Main Strengths:**

The paper is overall well written and introduces the necessary concepts. The result is solid and interesting.

**Q4 Main Weakness:**

In general, while the results are interesting, the contribution also feels a bit limited in scope and substance for UAI. The technological advances are non-trivial but also seem to be based on one crucial idea (the exp-sum-log trick). Some concepts in the paper are introduced but never referred to (smooth and decomposable circuits) and it is not clear how the hardware acceleration properties relate to complexity aspects (like decomposability) of PCs.

**Q5 Detailed Comments To The Authors:**

Introduction: the motivation for hardware realisation of PC is weak given the concerns raised on NNs. This needs either more unpacking or a different justification.
background line 6: exiting -> exact
page 4 P2 line 4 -> integer

**Q9 Complying With Reviewing Instructions:**

Yes

---

> ### Author Rebuttal · Authors · 2024-04-04
>
> We thank the reviewer for the constructive feedback, and for acknowledging that our technical contribution is non-trivial.
>
> Regarding the smooth and decomposability concepts:
> Smoothness and decomposability are structural properties of PCs required for tractable computation of marginal probabilities. Determinism is an additional property required for tractable MAP queries. In this work, all PCs are considered smooth and decomposable and we present additional theoretical results for deterministic PCs. We will improve the clarity in the experiments section to highlight the use of smooth and decomposable PCs.
> Our proposal for hardware acceleration utilizes those structural properties to perform error correction, as highlighted in Section 4.
>
> Answer to Q5:
> Thank you for your comments about the justification for using hardware PCs. We will try to clarify our views here.
> Probabilistic reasoning (possibly in conjunction with NNs) on edge devices is crucial. Especially on edge devices reasoning has to be performed efficiently and under uncertainties as, e.g., sensor data is exposed to various forms of noise or corruptions, and decisions need to be performed with low latency and low energy costs. We will improve the motivation of the paper to better highlight the need for hardware acceleration of PCs, which enables efficient probabilistic reasoning.
>
> We thank the reviewer for pointing out typos.

---

### Official Review · Reviewer_6SzF · 2024-03-27

**Q2-1 Originality-Novelty:** 3
**Q2-2 Correctness-Technical Quality:** 3
**Q2-5 Clarity Of Writing:** 4

**Q1 Summary And Contributions:**

The work introduces a dedicated approximate computing framework for energy-efficient inference in probabilistic circuits on hardware.

**Q2-3 Extent To Which Claims Are Supported By Evidence:**

4: Excellent: all claims are supported by very convincing evidence (in the form of comprehensive experimental evaluation, rigorous mathematical proofs, detailed (pseudo-)code, precise references, well-motivated and realistic assumptions) and the authors deliver what they promise.

**Q2-4 Reproducibility:**

2: Fair: key resources (e.g. proofs, code, data) are unavailable but key details (e.g. proof sketches, experimental setup) are sufficiently well-described for an expert to confidently reproduce the main results.

**Q3 Main Strengths:**

- originality
- technical quality

**Q4 Main Weakness:**

--

**Q5 Detailed Comments To The Authors:**

It's not clear to me the distinction between MAR and MAP

what kind of unit measure is used for the Energy column in table 1?

**Q9 Complying With Reviewing Instructions:**

Yes

---

> ### Author Rebuttal · Authors · 2024-04-04
>
> We thank the reviewer for recognizing the importance of our work and for appreciating the experiment results.
>
> Regarding the distinction between MAR and MAP:
> For evidence or marginal queries (MAR), we compute the (marginal) probability value while maximum-a-posteriori (MAP) queries compute the most probable path (sometimes called MPE). Therefore, we use different accuracy evaluation metrics for them. We focus on the numerical accuracy of MAR, and the classification sensitivity of MAP.
>
> Regarding the unit measure used for the Energy column in Table 1:
> The energy is the cumulative energy of every multiplier in the model, and the power of each multiplier is simulated with 65nm CMOS technology with a unit of uW. This value is normalized regarding the baseline energy calculated for full precision floating point. We will clarify this point in the text.

---

### Meta-Review · Area_Chair_HTn3 · 2024-04-16

The paper introduces a framework of approximate computing for energy-efficient inference in probabilistic circuits. The basic idea is to replace floating-point multiplications by integer additions in the logarithmic domain (which in itself is an old idea), with an appropriate error correction routine.

The reviewers found the questions studied interesting and new in the context of probabilistic circuits, and the research well conducted and reported. That said, also mild concerns were raised regarding the novelty of the ideas and the significance of the results.